# TriTrypDB: An integrated functional genomics resource for kinetoplastida

**Achchuthan Shanmugasundram[1], David Starns[1], Ulrike Böhme[1], Beatrice Amos[1], Paul A. Wilkinson[1], Omar S. Harb[2], Susanne Warrenfeltz[3], Jessica C. Kissinger[3], Mary Ann McDowell[4], David S. Roos[2], Kathryn Crouch[5]\*, Andrew R. Jones[1]\***

**1** Department of Biochemistry and Systems Biology, Institute of Integrative, Systems and Molecular Biology, University of Liverpool, Liverpool, United Kingdom, **2** Department of Biology, University of Pennsylvania, Philadelphia, Pennsylvania, United States of America, **3** Center for Tropical & Emerging Global Diseases, Department of Genetics, Institute of Bioinformatics, University of Georgia, Athens, Georgia, United States of America, **4** Department of Biological Sciences, Eck Institute for Global Health, University of Notre Dame, Notre Dame, Indiana, United States of America, **5** School of Infection and Immunity, University of Glasgow, Glasgow, United Kingdom

\* Kathryn.Crouch@glasgow.ac.uk (KC); andrew.jones@liverpool.ac.uk (ARJ)

## Abstract

Parasitic diseases caused by kinetoplastid parasites are a burden to public health throughout tropical and subtropical regions of the world. TriTrypDB (https://tritrypdb.org) is a free online resource for data mining of genomic and functional data from these kinetoplastid parasites and is part of the VEuPathDB Bioinformatics Resource Center (https://veupathdb.org). As of release 59, TriTrypDB hosts 83 kinetoplastid genomes, nine of which, including *Trypanosoma brucei brucei* TREU927, *Trypanosoma cruzi* CL Brener and *Leishmania major* Friedlin, undergo manual curation by integrating information from scientific publications, high-throughput assays and user submitted comments. TriTrypDB also integrates transcriptomic, proteomic, epigenomic, population-level and isolate data, functional information from genome-wide RNAi knock-down and fluorescent tagging, and results from automated bioinformatics analysis pipelines. TriTrypDB offers a user-friendly web interface embedded with a genome browser, search strategy system and bioinformatics tools to support custom *in silico* experiments that leverage integrated data. A Galaxy workspace enables users to analyze their private data (e.g., RNA-sequencing, variant calling, etc.) and explore their results privately in the context of publicly available information in the database. The recent addition of an annotation platform based on Apollo enables users to provide both functional and structural changes that will appear as 'community annotations' immediately and, pending curatorial review, will be integrated into the official genome annotation.

## Author summary

Kinetoplastid parasites cause severe infections in humans including African sleeping sickness, Chagas disease and leishmaniasis, which are classified as neglected tropical diseases. With the advancement of sequencing technologies, more and more genome sequences are being generated and deposited in archival repositories. TriTrypDB (https://tritrypdb.org),

**Data Availability Statement:** All relevant data are within the paper and the supporting database, https://tritrypdb.org/. All our bioinformatics tools

and resources are maintained in the VEupathDB GitHub repository: https://github.com/VEuPathDB.

**Funding:** This work was supported by the National Institute of Allergy and Infectious Diseases (NIAID, https://www.niaid.nih.gov/) — National Institutes of Health (NIH) Contract #75N93019C00077 to DSR, MAM & JCK and Wellcome Trust (https://wellcome.org) grants 218288/Z/19/Z and 212929/Z/18/Z to ARJ, KC, DSR & JCK. The funders had no role in study design, data collection and analysis, decision to publish, or preparation of the manuscript.

**Competing interests:** The authors have declared that no competing interests exist.

a component database of the VEuPathDB resource (https://veupathdb.org) is a free data mining resource, which currently hosts a subset of these parasite genomes that are of clinical and scientific importance. TriTrypDB also integrates functional genome-scale datasets (e.g. transcript expression, protein expression, genetic variation data) and information predicted from automated bioinformatics pipelines and from manual curation. TriTrypDB provides a user-friendly web interface and a number of tools and functions for users to conduct *in silico* experiments to ask questions and generate hypotheses. Researchers can also contribute their expertise via the User Comments form and Apollo annotation platform, and utilize our cloud-based workspace to analyze their own data. TriTrypDB has been extensively used by the research community over the last decade and serves as a primary resource for communities working with these organisms.

## Introduction

TriTrypDB (https://tritrypdb.org) is a component database of the Eukaryotic Pathogen, Vector and Host Informatics Resource (VEuPathDB, https://veupathdb.org) Bioinformatics Resource Center [1] and is supported by the US National Institutes of Allergy and Infectious Diseases (NIAID) [2] and the Wellcome Trust UK. TriTrypDB is a free online resource for data mining and multi-omic analysis of Kinetoplastid parasites. Beyond TriTrypDB, VEuPathDB also provides resources for other eukaryotic protist parasites (Apicomplexa, Amoeba, Giardia, etc.), fungi (both pathogens and non pathogens, https://fungidb.org) [3], vectors (arthropods and molluscs, https://vectorbase.org) [4], selected mammalian host data (https://hostdb.org), orthology determination and phylogenetic inference (https://orthomcl.org) [5], and clinical epidemiological (https://clinepidb.org) [6] and microbiome data (https://microbiome.org) [7].

As of release 59 (30th August 2022), TriTrypDB hosts 83 genomes from 36 kinetoplastid species. Although TriTrypDB predominantly hosts genomes of *Leishmania* and *Trypanosoma* species, it also includes genomes of species belonging to other clades in the Kinetoplastea class, such as *Angomonas*, *Crithidia* and *Leptomonas*. TriTrypDB integrates a wide range of other data types, including transcriptomic, protein expression, epigenomic and genetic variation data, phenotypes and experimental cellular localization data from fluorescent tagging. These data are obtained either from repositories, such as International Nucleotide Sequence Database Collaboration (INSDC) for genome assemblies or Sequence Read Archive (SRA) for functional sequencing data, or directly from providers. Several data types are analyzed by TriTrypDB and made available for computational interrogation. Processing is carried out using standard workflows and integrated using an ontology-driven framework to ensure data comparability across studies.

TriTrypDB provides an easy-to-use web interface with record pages that compile all data for entities such as genes, genomic sequences, single nucleotide polymorphisms (SNPs) and metabolic pathways, a genome browser for visualization of sequence data, publicly available bioinformatics tools for data analysis, a search strategy system for interrogation of pre-analyzed data, and a private Galaxy workspace [8] for analyzing primary data and examining it in the context of public data already loaded into TriTrypDB. Expert knowledge from the user community is captured in the form of User Comments and community annotations via Apollo [9] and this information is reviewed and incorporated to improve gene models and functions. Here we present a general overview of the current state of TriTrypDB, highlighting the major developments in the last decade and since the initial publication of this resource in 2010 [10].

## Methods

### Data integration

**Integrated datasets.** TriTrypDB release 59 hosts 83 genomes and 176 other functional datasets relating to *Trypanosoma*, *Leishmania* and a few other kinetoplastid species. New datasets and functionality are incorporated into TriTrypDB via bimonthly releases. Integrated datasets can be found on the datasets page under the 'Data' menu and those integrated in a recent release can be found on the news section (https://tritrypdb.org/tritrypdb/app/static-content/TriTrypDB/news.html). An overall trend of datasets (both genomes and other functional datasets) integrated between release 1.0 and release 59 is illustrated in Fig A in S1 File.

Of these 83 genomes, 31 are from *Leishmania* and 43 are from *Trypanosoma* genera. TriTrypDB also has genomes of other parasitic kinetoplastids including, *Angomonas deanei [11]*, *Blechomonas ayalai*, *Crithidia fasciculata [12]*, *Endotrypanum monterogeii [12]*, *Leptomonas pyrrhocoris [13]*, *Leptomonas seymouri [14]*, *Paratrypanosoma confusum* [15], and *Porcisia hertigi [16,17]* and free-living nonparasitic kinetoplastid *Bodo saltans*. Gene annotations are available for 66 of these sequences, while the remaining 17 are genome assemblies without annotations (Table 1). Of these 66 genome sequences with annotations, 36 are classified as 'reference' (or 'representative') sequences representing distinct species, while the remaining 30 are either additional strains of the already existing reference species or resequencing of already available strains.

The other omics data types available in TriTrypDB are listed in Table 1, which includes single-cell RNA-Seq datasets from *T. brucei [19]* that were integrated for the first time in release

**Table 1. Summary of integrated datasets.**

|  | Dataset Categories | Number of datasets |
|---|---|---|
| Genomics | Genome sequences and annotations | 66 |
|  | Genome sequences only | 17 |
| Transcriptomics | Expressed sequence tags (ESTs) | 1[a] |
|  | RNA-Seq | 39 |
|  | Single-cell RNA-Seq[b] | 2 |
|  | Microarrays | 10 |
| Proteomics (MS-based) | Identification | 12 |
|  | Quantitative | 10 |
|  | Organellar | 16 |
|  | PTMs | 13 |
| Epigenomics | ChIP-ChIP | 2 |
|  | ChIP-Seq | 7 |
| Variation | Sequence variation | 28 |
|  | Array probes | 1 |
| Phenomics | Quantitative (high throughput RNAi-target sequencing) | 1 |
|  | Curated | 1 |
| Sequence sites, features and motifs | Origins of replication | 5 |
|  | BAC, PAC and Cosmid-end sequences | 6 |
|  | Maxicircle annotations | 2 |
|  | Additional gene predictions | 1 |
|  | Transcript alignments from previous annotations | 3 |

[a]ESTs are obtained from the dbEST database [18].
[b]This is a new data type integrated for the first time in release 59.

59. Other key data types integrated into TriTrypDB include cellular localization images, orthology profiles assigned with the OrthoMCL algorithm [5] and metabolic pathways. TriTrypDB hosts microscopy images and annotations from the TrypTag project, which aims to tag and determine the cellular localization of every protein encoded in *T. brucei* TREU927 genome [20]. Metabolic pathways are integrated from KEGG [21], MetaCyc [22], TrypanoCyc [23] and LeishCyc [24] and are represented using Cytoscape JS [25], an open source graph library.

Since release 59, TriTrypDB integrates protein structure predictions from AlphaFold [26,27], an artificial intelligence system created by DeepMind (https://www.deepmind.com/). The integrated AlphaFold predictions are the EMBL-EBI predictions, currently covering sequences from the UniProt reference proteome and these predictions will be automatically updated with new releases from AlphaFold in the future. These predictions are integrated in TriTrypDB by mapping TriTrypDB gene IDs to UniProt IDs when there is a corresponding entry in UniProt and by sequence similarity when TriTrypDB genes do not have an exact match in UniProt. These protein structures can be visualized via the 3D viewer in the gene pages. Gene or protein features are cross-referenced with external databases including Chemical Entities of Biological Interest (ChEBI) [28], Enzyme Nomenclature [29], Gene Ontology [30,31], PDB [32–34], IEDB [35] and NCBI Taxonomy [36] and information from these external resources are also integrated into TriTrypDB. These datasets can be downloaded or used with the site tools available under the 'Tools' menu.

**Updated data processing.** Genome sequences loaded into TriTrypDB are first obtained from an INSDC repository (https://www.insdc.org) and processed by European Bioinformatics Institute (EBI) workflows. During EBI processing, assembly core statistics are generated and DNA features are predicted using RepeatMasker (https://www.repeatmasker.org), Dust-Masker [37] and Tandem repeat finder [38]. Protein features are predicted with InterProScan [39], SignalP [40], TMHMM [41], Panther [42], and PSIPRED [43]. RNA features are predicted with Rfam [44], tRNAscan [45] and miRBase [46]. Genomic data from the EBI workflow are supplemented with data generated from in-house pipelines (https://github.com/VEuPathDB) including open reading frames (ORFs), EST alignments and synteny plots generated using Mercator and MAVID [47]. A small number of legacy genomes that were not submitted to INSDC repository by the genome providers are processed by in-house pipelines from VEuPathDB, although we no longer accept genome assemblies that are not submitted to an INSDC repository.

All DNA, RNA and protein datasets are aligned to their respective reference genome sequences. Alignments are used for downstream processing including variant calling, copy number variation analysis, and differential gene expression analysis. The EBI RNA-seq alignment pipeline uses HISAT2 [48] for alignment to the reference genome sequence and HTSeq-count [49] for counting reads aligned to genome features. This is followed by in-house processing to generate TPM-normalized data for plotting and fold-change queries, normalized bigwig files for visualization in JBrowse, and to run DESeq2 [50] analysis on all pairwise conditions for differential expression queries. SNP and copy number variation (CNV) analyses are conducted using VEuPathDB pipelines where Bowtie2 [51], Samtools [52] and VarScan [53] are utilized for calling SNPs and normalized coverage data is used to predict chromosome and gene-scale copy number variations. Functional data such as gene names and product descriptions are assigned using the Ensembl Xref pipeline that links functional annotation for proteins analyzed by Uniprot. Large scale parallel compute is conducted by EBI and the generated data are loaded into a relational database at VEuPathDB. More details on the architecture of the database and data processing pipelines can be found in Amos *et al*. [1].

## Results

### A user-friendly web interface

The VEuPathDB portal and all component sites including the TriTrypDB database share the same backend infrastructure and web interface. The VEuPathDB user interface is continually improved to provide convenient and consistent access to data, searches, tools and help information.

**Homepage.** The homepage (Fig 1) features a header that is present on all pages, a main panel, an expandable 'News & Tweets' section on the right, and a footer with clickable icons to access other VEuPathDB resources (Fig 1G). The center of the header includes a site search

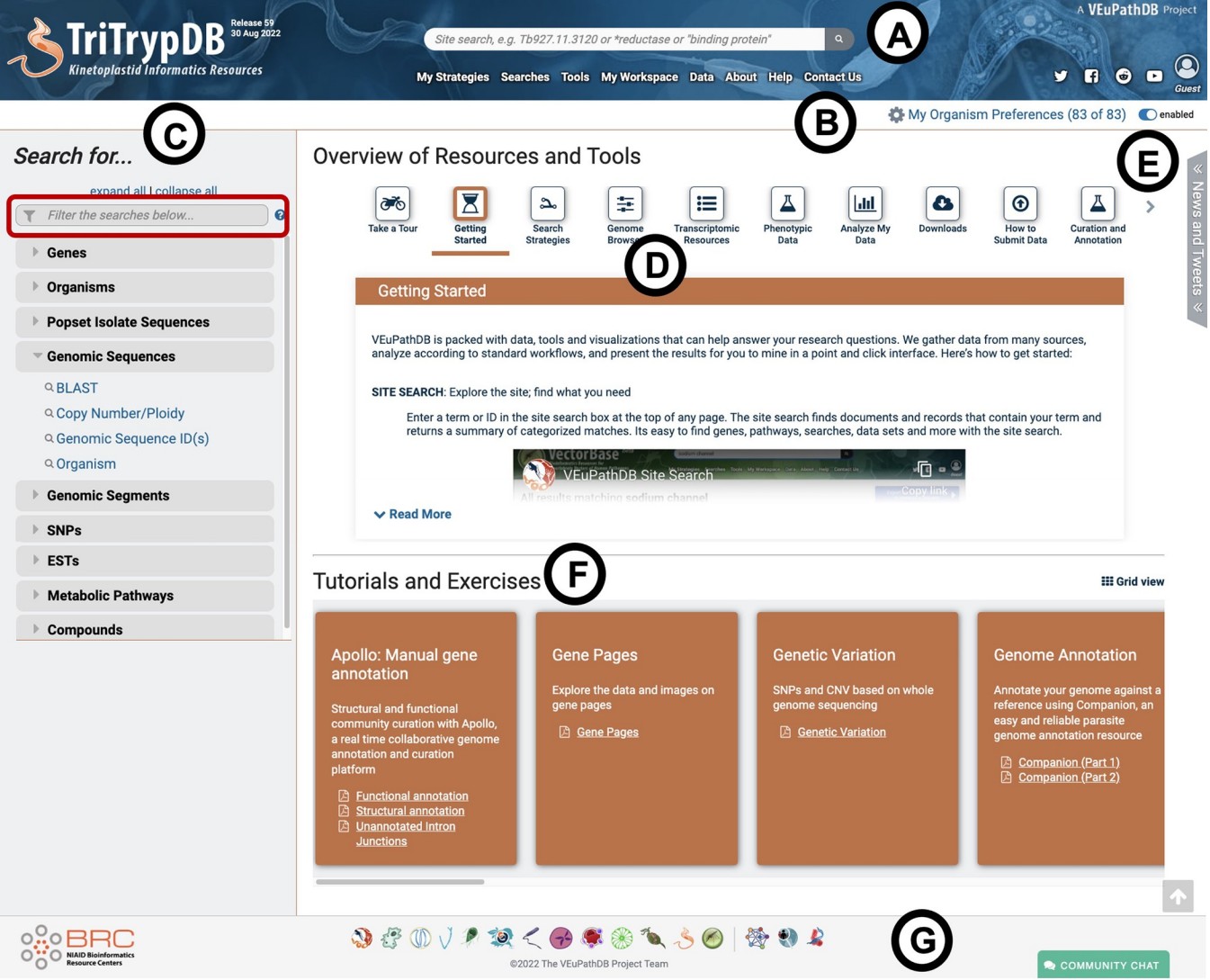

**Fig 1. Home page of TriTrypDB.** (A) Header on all site pages that includes site search, menu bar providing access to all searches, data and tools, and links for social media, registration, login and user profile. (B) The recently implemented 'My Organism Preferences' filter. (C) The left hand search panel contains searches of different data in the database, organized into categories. (D) The overview of resources and tools section provides vignettes to help users get started on specific tools and resources of interest. (E) The expandable news and tweets section (collapsed by default) offers quick access to news releases and recent tweets. (F) Links to more detailed step-by-step instructional material. (G) Footer consisting of hyperlinked logos to other VEuPathDB resources in addition to the Gitter Community Chat button.

box (see below) and a 'menu' bar that appears below the site search that provides quick access to 'My Strategies', 'Searches', 'Tools', 'My Workspace', 'Data', 'About', 'Help' and 'Contact Us' sections (Fig 1A). Social media, login, registration and user profile links are displayed on the right corner of the header.

A newly implemented feature named 'My Organism Preferences' is found below the right corner of the header on all pages (Fig 1B). This function offers the users the option to filter menus to a subset of organisms of their interest. TriTrypDB will therefore function as a personalized database containing only their chosen organisms. This feature can easily be enabled or disabled at any time from the button located next to the 'My Organism Preferences' on any page.

The left sidebar of the main panel of the home page provides access to all available searches from the database (Fig 1C). Searches are organized into expandable categories and the users can also refine the list of searches based on key words using the filter above the menu (Fig 1C, red box). The top of the central portion of the main panel displays a list of scrollable vignette buttons, which when clicked provide useful information on various tools and resources available in TriTrypDB (Fig 1D). A section on 'Tutorials and Exercises' is displayed below the vignettes, which provides access to step-by-step tutorials downloadable in PDF format (Fig 1F). An expandable section on 'News and Tweets' to the right of the vignettes provides a quick exploration of the website news and recent tweets (Fig 1E).

**Record pages.**   Gene records compile all the available information about a particular gene into a single web page. The information about a gene and its function that are available in the gene page are generated from integrated datasets and automated pipelines (See 'Data integration' above). These record pages are updated on a regular basis both to provide better user experience and to ensure the relevance of content displayed in the record pages.

TriTrypDB gene record pages can be navigated via the thumbnail 'Shortcuts' at the top and the collapsible 'Contents' menu on the left. The data are categorized into 19 different categories as displayed in the 'Contents' menu. Help icons (blue question mark icons) are available at multiple locations in the gene pages to help users find additional information where it is appropriate. Gene pages host a multitude of data including gene model information, functional annotation (see 'Functional annotation' section below), ortholog and paralog predictions from the OrthoMCL pipeline [5], experimental data including transcript expression, protein expression and phenotyping data (see 'Integrated datasets' section above), immune epitopes from IEDB database [35], link outs to external databases such as UniProtKB [54], TrypsNetDB [55] and PDB [32–34], link outs to relevant literature from PubMed, protein structure predictions from AlphaFold [26,27], and predictions of protein features and properties such as molecular weight, InterPro motifs [56], signal peptides and transmembrane domains.

TriTrypDB provides different types of data representations (e.g. tables, bar charts and plots) to facilitate better exploration and visualization of omics data on the gene pages, including a summary graph (using Plotly, https://plotly.com) of a gene's expression values across all integrated RNA-Seq datasets. This representation is useful for observing overall trends in expression of a gene across experiments and to identify outliers that may require further exploration. Another example is CELLxGENE (https://cellxgene.cziscience.com), a single-cell visualization platform developed by the Chan-Zuckerberg initiative, which was implemented in release 59 to facilitate exploration of single-cell RNA-Seq data. TriTrypDB gene pages provide static images and links to explore the data dynamically in the CELLxGENE platform, enabling the user to select or paint groups of cells based on gene expression, experimentally derived metadata such as clusters or metadata derived from the experimental design such as condition or replicate. An example dataset can be viewed at https://tritrypdb.org/cellxgene/view/ tbruTREU927_briggs_wt_cellxgene_RSRC.h5ad/.

Similar to gene record pages, TriTrypDB also provides pages for other record types including popset isolate sequences, genomic sequences (scaffolds), genomic segments, SNPs, ESTs, metabolic pathways and chemical compounds. These record pages can be accessed by conducting searches (see Search strategy system section below) for the respective entity types other than genes (e.g. popset isolate sequences) and accessing the record IDs for the retrieved entities (Fig B in S1 File). Several of these record types (e.g. metabolic pathways) can also be accessed from the gene record pages via clicking links for the record IDs under the respective sections in gene pages. The record pages of these additional entity types are also organized in a similar fashion to gene pages with the data organized into multiple categories and the displayed data can be navigated via the collapsible content menu on the left.

## Tools

**Site search.**    The search bar present in the header of all TriTrypDB web pages (Fig 1A) allows users to perform a site-wide search with gene identifiers and free text. This search returns a categorized list of results with filters available for users to define categories or organisms of interest. The site search results include pages of datasets, news items and tutorials in addition to feature record pages (genes, pathways etc.). The site search results corresponding to records can be exported as a step in the search strategy system allowing further data exploration and download.

**Search strategy system.**    The search strategy system available in TriTrypDB provides a unique and powerful mechanism to mine the vast amount of omics datasets and to integrate results in a multi-step '*in silico*' experiment (Fig 2). Multi-step search strategies are built one step at a time, choosing the first search from either the 'Search for. . .' on the home page (Fig 1C) or the 'Searches' menu on the header (Fig 1A). Searches are also available for other feature types (listed above in the 'Record pages' section) such as genomic segments, SNPs and pathways.

Search results are displayed in a newly designed 'My Search Strategies' page. The top graphic panel on this page displays growing search strategies (Fig 2A), and these search strategies can be extended by clicking 'Add a step' (Fig 2A, red box). The options for adding steps include 'Combine' for use with similar records using union, intersection and minus operations, 'Transform' to find records cross-referenced to the current results (orthologs, metabolic pathways and compounds) and genomic colocation searches to find features that are related by their location in the genome (genes, genomic segments and SNPs) (Fig 2C). Search strategies can be made, copied, edited and deleted by any user. Users with a free account can additionally save strategies and share them with others using a private link. The users can add relevant details to the description section when saving a strategy, which will help them remember key details when they access it at a later time point. Saving a strategy retains the order of steps and parameter values, but not the actual results as subsequent database versions containing new data may alter the results.

The results table appears below the graphic panel (Fig 2B) and it includes a list of resulting feature IDs and associated data. Columns of associated data can be added to the results table via 'Add Columns' and results can be downloaded locally using the 'Download' option (Fig 2B, red box). The recently implemented 'Send To' dropdown menu (Fig 2B, red box) allows users to save their search results as an ID list in the 'My Data Sets' page, as a downloadable text file for future analyses, or to transfer the list from TriTrypDB to VEuPathDB and analyze in the context of all VEuPathDB organisms. By saving both search strategies and search results from those strategies, users can compare results from different versions of the database in the future. The collapsible organism filter that appears on the left of the results table (Fig 2B, red arrow)

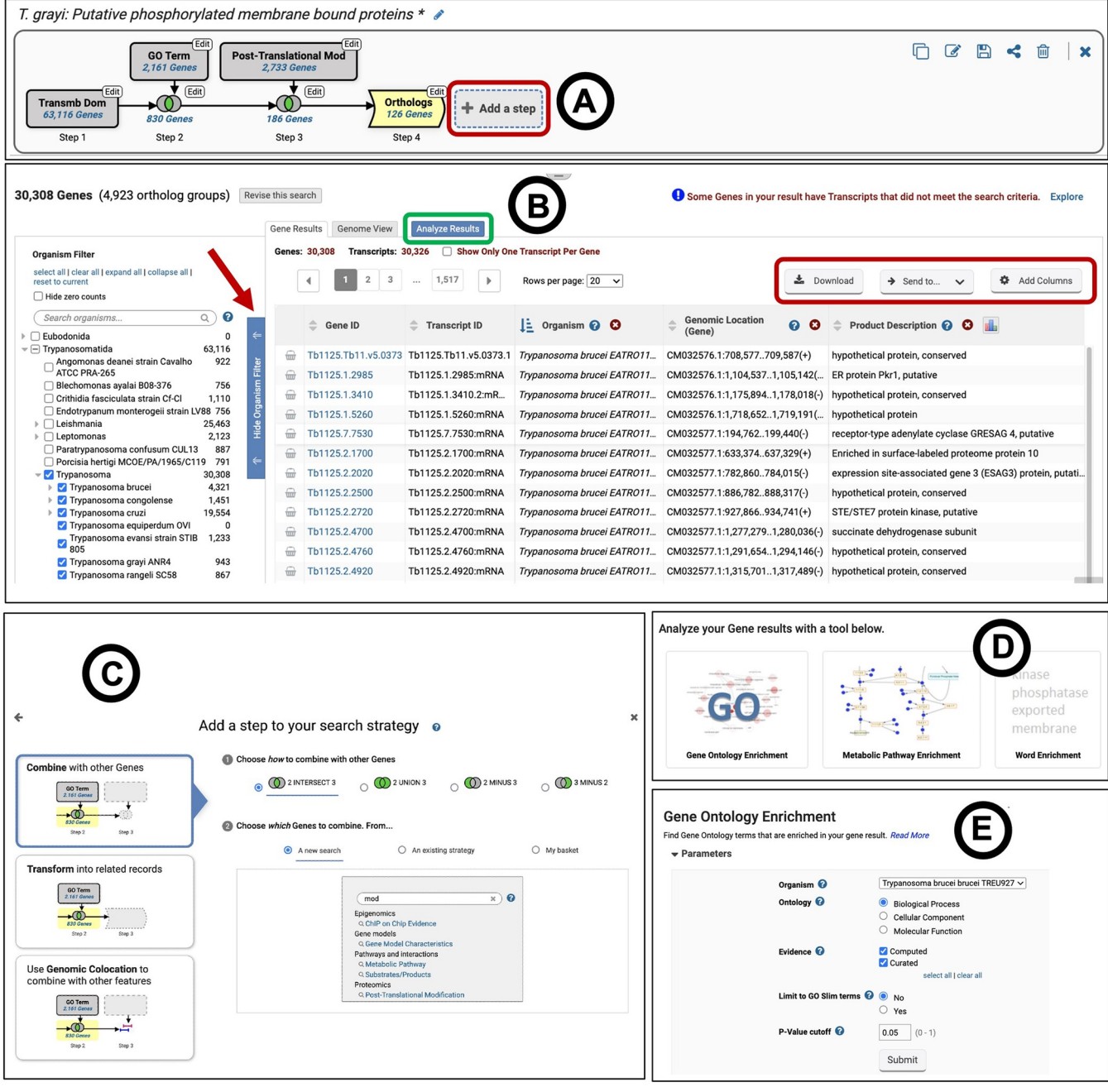

**Fig 2. Search strategies as *in silico* experiments and functional enrichment analysis.** (A) The graphic panel shows an example multi-step search strategy (https://tritrypdb.org/tritrypdb/app/workspace/strategies/import/9c17640460e66cd5). (B) Results of the first step of the search strategy, with the redesigned vertical organism filter on the left (red arrow) and the results table on right. (C) Redesigned 'Add Step' popup showing the three options to add steps, 'Combine', 'Transform' and 'Genomic Colocation', and the details of the chosen option. (D) The three options available for the analysis of gene results: GO enrichment, metabolic pathway enrichment and word cloud. (E) The form for selecting appropriate parameters for GO enrichment analysis.

shows the distribution of the results across the organisms searched and the results can be limited to organisms of interest using the filter.

**Enrichment analysis.** TriTrypDB provides tools for users to perform functional enrichment of gene results arising from search strategies or user-supplied gene lists (Fig 2B, highlighted in green box). Available tools include word, gene ontology (GO) [30,31] and

metabolic pathway enrichment analyses (Fig 2D). These tools use Fisher's exact test in combination with multiple test corrections implemented using both Bonferroni and Benjamini-Hochberg methodologies. Simple word enrichment is useful in detecting keywords that are enriched in annotations such as product descriptions, user comments and annotator notes when more formal GO and metabolic pathway annotations are not available for a gene. An option to limit the GO terms to the slim subset is available with the GO enrichment analysis to reduce the redundancy of enriched terms (Fig 2E). GO enrichment results can also be exported to REVIGO [57], which facilitates data visualization via a variety of interactive tools. For metabolic pathway enrichment analysis, TriTrypDB provides an option to choose pathways from KEGG [21] and MetaCyc [22].

**Genome and protein browsers.** TriTrypDB currently embeds the JBrowse genome browser [58] to facilitate the dynamic visualization of annotations and functional data on genome sequences. JBrowse is an open source and configurable platform that offers improved browsing and zooming speed and the ability to save and share personalized views. JBrowse enables users to select and display tracks with aligned transcriptomic, proteomic, epigenomic and variation data on the genome browser. Transmembrane domains (TMHMM predictions [41]), protein domains from InterPro [56] and synteny views across multiple genomes can be accessed via the protein browser.

**Sequence analysis tools.** The protein features and properties section of gene record pages currently provides direct access to six external bioinformatics tools for analysis of protein sequences. These are BLAST-P [59,60], InterProScan [39], big-PI predictor [61], MitoProt [62], STRING [63] and WoLF PSORT [64]. The users can submit a protein sequence for analysis by clicking a tool's 'Submit' button from the record page of the relevant gene, which opens a new tab and initiates the respective query on the external web interface of the chosen tool. MitoProt, WoLF PSORT and big-PI predictor are used for the prediction of mitochondrial targeting signal, subcellular localization site and glycosylphosphatidylinositol anchor respectively. STRING accesses visualizations of both known and predicted protein-protein interactions.

The Clustal Omega [65] tool is embedded in the orthologs and paralogs section of record pages. Users can align a gene sequence with the sequences of its homologs by selecting the genes from 'Orthologs and Paralogs within TriTrypDB' table, choosing the sequence type and the output format, and clicking the 'Run Clustal Omega for selected genes' button.

**Galaxy interface.** The VEuPathDB Galaxy [8] interface offers an environment for users to privately analyze their own data as well as data available at INSDC repositories. Users can either upload their own data or use the sample accession numbers from European Nucleotide Archive (ENA) or Sequence Read Archive (SRA) for data transfer into Galaxy. Individual samples can be placed into dataset collections for efficient organization and downstream processing. Currently, TriTrypDB provides preconfigured workflows for RNA-seq data (for identification of transcript expression from single and paired end stranded and non-stranded Illumina data and for differential expression analysis), variant calling and for mapping proteins to ortholog groups using the OrthoMCL algorithm [5]. There are a host of tools that are available to use individually or for users to create their own workflows using the built-in user interface. Analysis results can either be downloaded or exported into TriTrypDB for private exploration through custom searches, the search strategy system or genome browser visualizations.

## Curation and annotation

Nine selected kinetoplastid genomes (four *Leishmania* and five *Trypanosoma*) that were sequenced by the Parasite Genomics group at Wellcome Sanger Institute and previously

hosted in GeneDB [66] were manually curated by expert curators from the VEuPathDB project by utilizing the curation infrastructure from the Wellcome Sanger Institute. This joint effort between GeneDB and TriTrypDB in improving these parasite genomes was made possible by continuous funding from the Wellcome Trust for the TriTrypDB database since 2012. The updated annotations are regularly integrated into TriTrypDB through bimonthly releases. As GeneDB was taken offline last year, TriTrypDB serves as the sole authoritative resource for the annotation of these genomes. TriTrypDB continues to curate these genomes and functional annotations were updated in releases from the last year.

Although structural and functional annotations are updated by VEuPathDB, improving genome assemblies with new data is beyond the scope of this project. However, VEuPathDB sources new and improved assemblies of already integrated genomes from INSDC repositories as detailed above. These new genomes can either replace already existing genomes or be integrated as additional genomes of already existing species/ strains in TriTrypDB, depending on the data quality or perceived community importance. VEuPathDB has also obtained permissions from the majority of the genome data providers to update annotations in INSDC repositories. Genomes with significant annotation updates (either from VEuPathDB curation or from community curation via Apollo) are planned to be updated in INSDC repositories in the future.

**Structural annotation.** Re-annotation efforts have resulted in sequence changes and gene model updates for the nine curated kinetoplastid genomes. These annotation changes include creation of new genes and obsoletion of existing genes including pseudogenes, addition of alternative transcripts, changes to existing gene models, transcripts and coding sequences, and changes to gene IDs (Table 2). The gene IDs remain stable despite changes to gene models in the majority of cases and the gene IDs are not versioned with structural annotation updates. However, all changes are tracked in the curation database and the changes are documented in the annotation logs associated with the news of our bimonthly releases. TriTrypDB also stores previous identifiers of genes with new gene IDs in the database and users can access the record pages of these genes with new IDs by searching with their previous IDs. In addition, putative gene models removed from official annotations due to lack of evidence for them to be real are made obsolete rather than being deleted and they can be reverted back and added to official annotations when new evidence becomes available. In addition all sequences and annotations from previous releases are made available in the downloads section and users can compare different versions of annotations and conduct analyses with these previous genome versions available in downloads, if required.

VEuPathDB annotation efforts are focussed on improving the structural annotations of these curated genomes in order to provide high quality data for these key parasite species. We

**Table 2. Summary of structural annotation changes for curated organisms since the initial integration of genomes in TriTrypDB.**

| Organism | Coding genes | | | Pseudogenes | | Changed Gene IDs | Total genes with updates |
|---|---|---|---|---|---|---|---|
| | New | Deleted | Changed | New | Deleted | | |
| *Trypanosoma brucei* TREU927 | 405 | 568 | 613 | 165 | 5 | 121 | 1877 |
| *Trypanosoma brucei* gambiense | 0 | 1894 | 88 | 74 | 0 | 2 | 2058 |
| *Trypanosoma congolense* | 0 | 1556 | 68 | 85 | 0 | 3 | 1712 |
| *Trypanosoma cruzi* | 2 | 0 | 156 | 7 | 8 | 9 | 182 |
| *Trypanosoma vivax* | 1 | 462 | 324 | 340 | 0 | 2 | 1129 |
| *Leishmania braziliensis* | 49 | 1 | 25 | 16 | 18 | 4 | 113 |
| *Leishmania donovani* | 5 | 3 | 58 | 4 | 0 | 57 | 127 |
| *Leishmania major* | 120 | 1 | 1474 | 4 | 6 | 1 | 1606 |
| *Leishmania mexicana* | 14 | 2 | 12 | 4 | 0 | 2 | 34 |

rely on our data providers to supply genome assemblies and annotations. The improvement of genome assemblies and generation of first-pass genome annotations for genome assemblies without annotations are beyond the remit of the VEuPathDB project. However, VEuPathDB curators have also generated first-pass structural annotations for genomes relevant to our user community, when requested by them. An example is the *Leishmania amazonensis* genome [67], which was annotated by the VEuPathDB project using the Companion genome annotation pipeline (https://companion.ac.uk/) [68], an external tool that can be accessed from the 'Tools' section of TriTrypDB.

**Functional annotation.** Functional annotation dominates the curatorial efforts undertaken by the TriTrypDB project. Functional annotation attributes include gene names/ symbols, synonyms, product descriptions, GO annotations, EC numbers, annotator notes, literature citations, previous database identifiers and external database references. Between releases 1.0 and 59, a total of 79,242 genes had functional annotation updates and these include 8,334 gene names and synonyms, 37,995 product descriptions and 92,041 GO annotations. A summary of complete functional annotation changes made over the last decade can be found in Table 3.

GO annotations are curated using the standards developed by the Gene Ontology (GO) Consortium and the curated GO annotations are accompanied by relevant metadata such as evidence codes (http://geneontology.org/docs/guide-go-evidence-codes/), references (PubMed IDs) and additional evidence in support of annotations (with/ from). In addition to these internally curated GO annotations, TriTrypDB also hosts electronically annotated GO annotations from InterPro2GO and UniProt and curated GO annotations from the TrypTag project. These metadata and source of annotations (e.g. GeneDB, InterPro, TrypTag) are displayed in the GO annotations section of gene pages (Fig C in S1 File) and in the GO annotation (GAF) files in the downloads section to help users to understand the methods used to assign different GO terms and to let them decide on whether to trust an annotation.

TriTrypDB collaborates with the Gene Ontology Consortium to create new terms to represent kinetoplastid biology, particularly cellular components and biological processes. A few examples of these newly obtained biological processes include acidocalcisome organization (GO:0106117), ciliary basal body segregation (GO:0120312), procyclogenesis (GO:0120324) and kinetoplast DNA replication (GO:0140909), and cellular components include flagellum attachment zone (GO:0120119), reservosome (GO:0106123), ciliary microtubule quartet (GO:0120260) and ciliary centrin arm (GO:0120269).

**Table 3. Summary of functional annotation updates for curated organisms since the initial integration of genomes in TriTrypDB.**

| Organism | Gene products | Names/ synonyms | GO annotations | EC numbers | PubMed IDs | Annotator notes | Total genes with annotations |
|---|---|---|---|---|---|---|---|
| *Trypanosoma brucei* TREU927 | 4525 | 2140 | 14565 | 344 | 6687 | 15242 | 9547 |
| *Trypanosoma brucei* gambiense | 2175 | 719 | 8326 | 76 | 109 | 241 | 7346 |
| *Trypanosoma congolense* | 8724 | 577 | 7041 | 81 | 71 | 260 | 11322 |
| *Trypanosoma cruzi* | 6733 | 1599 | 19629 | 333 | 3822 | 3939 | 20304 |
| *Trypanosoma vivax* | 1935 | 574 | 7424 | 67 | 49 | 208 | 5454 |
| *Leishmania braziliensis* | 2777 | 556 | 7498 | 154 | 620 | 530 | 4990 |
| *Leishmania donovani* | 2632 | 580 | 6732 | 70 | 67 | 208 | 4885 |
| *Leishmania major* | 3059 | 767 | 8215 | 167 | 926 | 1119 | 6069 |
| *Leishmania mexicana* | 2679 | 729 | 7165 | 75 | 334 | 297 | 5098 |

### Harnessing community expertise for genome annotation

The generation of genome sequences has been expanding at a scale larger than ever before and the number of kinetoplastid genomes in TriTrypDB has increased from five in release 1.0 to 83 in release 59. As only a limited number of genomes are curated by staff curators at VEuPathDB and the level of curation is much lower than that of model organism databases (MODs) even for the curated species, we offer User Comments and the Apollo web-based genome annotation platform [9] for research communities to contribute their expertise to improve annotations.

**User Comments.** User Comments offer the fastest way to add information to gene records and to alert the curation team in the case of curated genomes. TriTrypDB strongly encourages the user community to offer their expertise by submitting comments about new findings or publications, or even negative results. A new comment can be added by clicking on the 'Add a comment' link available on all gene record pages (Fig 3A). The submission of these comments requires users to create an account with TriTrypDB. Users can enter descriptive information about gene structure or function, upload a reference and files (e.g. images of protein localization), and add other gene identifiers (Fig 3B). All comments become immediately visible on the gene pages, searchable via either the site search or the text search from the menu, and can be modified or deleted at any time by the same user.

TriTrypDB currently (as of 30th August 2022) has 5,079 user submitted comments covering a total of 10,038 genes from 51 annotated genomes. Of these, 4,793 comments are associated

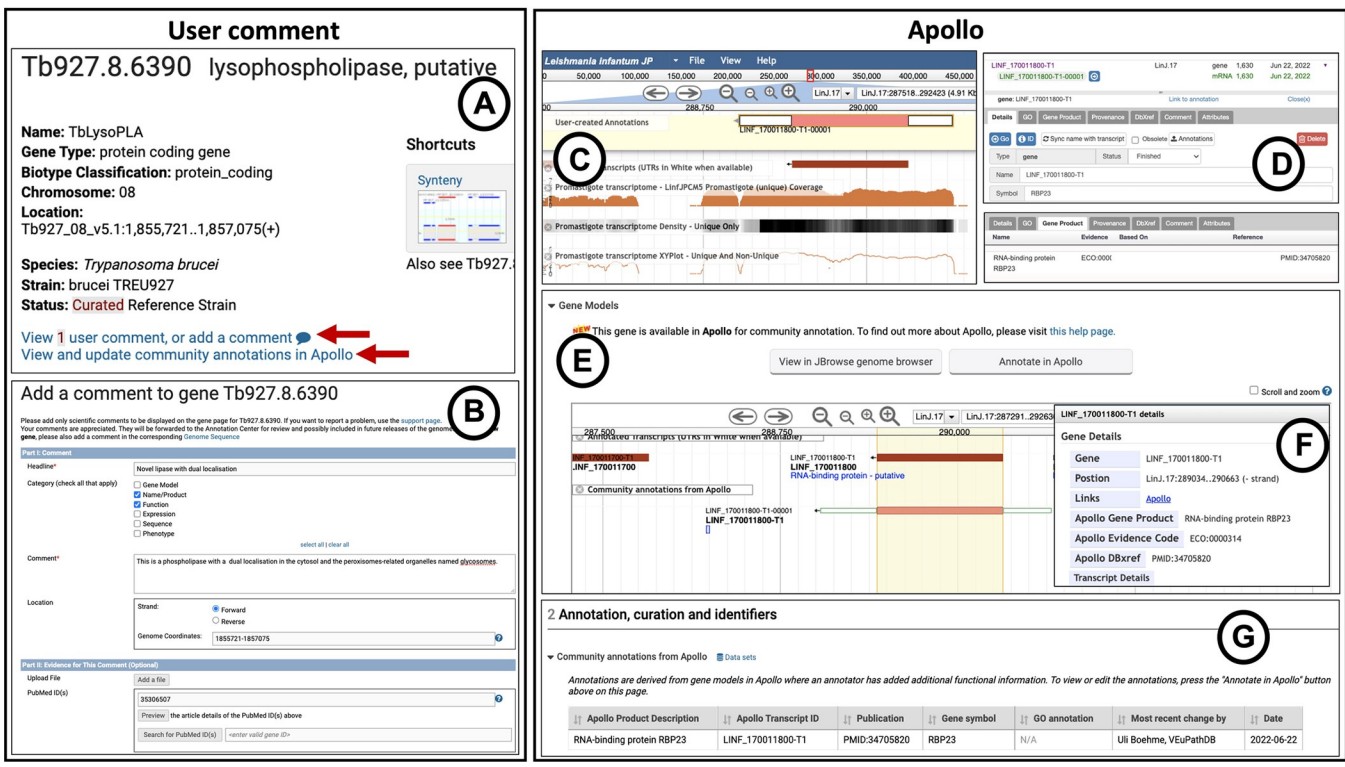

**Fig 3. Community curation via User Comments and Apollo.** (A) On the top of the gene record page are links to add User Comments and to access Apollo. (B) User Comment interface. (C) To annotate a gene in Apollo, the current gene model needs to be added to the User-created Annotations Area. (D) Interface in Apollo to add gene symbols, product descriptions, GO terms, database references and comments. Once the annotation is finalized, the status needs to be set as finished. (E) Finished annotations are shown on the gene page in the Gene models graphic in the track 'Community annotations from Apollo'. (F) A popup shows Apollo product descriptions, evidence code and the publication that is associated with the gene. (G) Apollo product description, gene symbol and publication are shown in the Annotation, curation and identifiers section on the gene record page.

with 9,320 genes from the nine currently curated genomes. Approximately 94% (4,506) of these comments from curated genomes have now been integrated into official annotations, while another 97 (~2%) comments have been reviewed, but not integrated into TriTrypDB.

**Community annotation via Apollo.**   TriTrypDB facilitates manual curation by community experts via Apollo, a collaborative genome annotation and curation platform [9]. This tool allows the editing of existing structural annotations and the creation of new annotations. Users can also update product descriptions and add other functional annotation attributes such as gene names/ symbols, GO terms, and publications associated with the gene. All TriTrypDB reference genomes and selected non-reference genomes (39 in release 59) are available in Apollo and can be accessed via the tools menu and links on gene record pages (Fig 3A). The only prerequisite to access Apollo is to create a TriTrypDB account and to log into Tri-TrypDB, as is the case with User Comments.

To initiate an annotation in Apollo, the gene model needs to be added into the User-created Annotations area. This can either be done by dragging and dropping or alternatively by using the right-click Apollo menu. Once the gene model is in the Annotation area, it can be modified based on evidence (Fig 3C) and functional annotations such as product descriptions can be added (Fig 3D). Users can also add comments, database references and literature references (PubMed IDs) in the annotation editor window in order to provide supporting evidence for structural or functional annotations made by them. By setting the status to finished, the curator indicates that the annotation is complete. Finished genes are represented in the community annotations track on the genome browser embedded under 'Gene Models' section of gene pages on the following day (Fig 3E and 3F). Similarly, added product descriptions can be found under 'Community annotations from Apollo' in the 'Annotation, curation and identifiers' section of gene pages (Fig 3G). These community annotations entered in Apollo are now indexed and are available for search via the site search. After review by VEuPathDB curators, community annotations are periodically integrated into the official gene set.

## Discussion

### Recent science enabled by TriTrypDB

The availability of the TriTrypDB resource has supported the advancement of kinetoplastid science, both in basic discovery and translational research over the last decade. Below are a few recent examples of how TriTrypDB data and tools have been utilized by the research community to conduct their own research. TriTrypDB genomes and annotations have been used to characterize individual genes [69–73] or gene families [74,75], identify orthologs across species [55,70,76], conduct genome-wide analyses to study genetic variations such as SNPs, CNVs and hybridization events [77–79], perform comparative genomic [80–82] and phylogenetic/ phylogenomic [82,83] analyses, and as reference genomes for the assembly and/or annotation of newly generated genomes [16,84]. Similarly, the genome assemblies and annotations have also been utilized for the analysis of differential gene expression [19,72,85], protein expression [85–87], the identification of post-translational modifications [72] and potential new genes missing in the official gene sets [88]. Genome data in TriTrypDB has also been instrumental in the development of other kinetoplastid-specific database resources such as TrypsNetDB [55] and TrypTag.org [20]. Moreover, TriTrypDB gene pages are hyperlinked from the record pages of external databases including UniProtKB [54], TrypanoCyc [23] and TDR Targets [89].

Some of the TriTrypDB tools that are popularly used by researchers include the genome browser [83,90], GO enrichment [19,86] and BLAST [71,79]. In addition, researchers have utilized integrated datasets including MS-based proteomics [91,92], RNAi [87] and SNPs [93], implemented searches ranging from gene IDs [94], text/ keywords [76,94], EC numbers [95],

protein features and properties such as InterPro domains, signal peptides and transmembrane domains [86,96,97], and the search strategy system [86] to ask their research questions or test their hypotheses. TriTrypDB serves as an invaluable tool for the selection of drug targets as reviewed in Osorio-Méndez *et al*. [98] and some of the studies discussed above have illustrated the role of TriTrypDB in the identification of potential targets for the design of drugs [73,80] and vaccines [96,97].

## Future perspectives

TriTrypDB will continue to add and improve tools, functional datasets and genome annotations in the future including functionality for integrated analysis and visualization of host-parasite interactions. Two host-parasite RNA-Seq datasets investigating the responses of *T. cruzi* infections on human cell lines [99,100] have already been integrated in VEuPathDB, with human data in HostDB (https://hostdb.org) and the corresponding parasite data in TriTrypDB. Currently, the search strategy system can be used for the separate interrogation of human and parasite data on respective databases, future development will include functionality to explore both host and parasite data in the context of one another. Maxi-circle sequences from *Leishmania* have also been loaded onto TriTrypDB and can currently be visualized via the genome browser; complete integration of these organellar genome sequences and development of appropriate gene record pages and tools for exploration of these data via search strategies is planned for the future.

TriTrypDB's future development plans include infrastructure for the functional integration of MS-based metabolomics data with sequence-based information, and tools for visual representation of different types of phenotypic information, loss-of-heterozygosity and haplotype data. A significant forthcoming challenge will be the integration of phased genomes, including tools for exploration of structural and sequence differences between haplotypes. With at least one kinetoplastid example already published [101] and others in progress, we anticipate that this will be a priority for the TriTrypDB community. VEuPathDB's efforts are also focused on synchronization of genome sequences and annotations with the INSDC data repositories, rolling out dedicated outreach activities for Apollo-based community annotations, development of a gateway resource for integrated exploration of VEuPathDB and bacterial/ viral BRC (https://www.bv-brc.org) and additional workflows for users to analyze their data with the Galaxy workspace.

## Supporting information

**S1 File. Additional supplementary figures. Fig A** Overall trend of genomes and other functional datasets available in TriTrypDB between release 1.0 (October 2009) and release 59 (October 2022). **Fig B** Accessing record pages of popset isolate sequences by conducting a dedicated search from the home page. **Fig C The Gene Ontology terms table from the gene pages.** An example from gene Tb927.8.4470 (chaperone protein DnaJ, putative, J40) showcasing annotations from multiple sources such as GeneDB, UniProt and TrypTag databases. The descriptions of data available in the different columns of this GO terms table are also provided here.
(DOCX)

## Acknowledgments

We would like to acknowledge the support of all the VEuPathDB team, including past and present team members (https://tritrypdb.org/tritrypdb/app/static-content/personnel.html)

and all members of the GeneDB team. We also wish to thank members of TriTrypDB research communities for their willingness to share their data, often prior to publication and for their continuous feedback to improve the resource.

## Author Contributions

**Conceptualization:** Achchuthan Shanmugasundram, Jessica C. Kissinger, Mary Ann McDowell, David S. Roos, Kathryn Crouch, Andrew R. Jones.

**Funding acquisition:** Jessica C. Kissinger, Mary Ann McDowell, David S. Roos, Kathryn Crouch, Andrew R. Jones.

**Project administration:** Mary Ann McDowell, David S. Roos, Kathryn Crouch, Andrew R. Jones.

**Writing – original draft:** Achchuthan Shanmugasundram, David Starns, Ulrike Böhme.

**Writing – review & editing:** Achchuthan Shanmugasundram, David Starns, Ulrike Böhme, Beatrice Amos, Paul A. Wilkinson, Omar S. Harb, Susanne Warrenfeltz, Jessica C. Kissinger, Mary Ann McDowell, David S. Roos, Kathryn Crouch, Andrew R. Jones.

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
