## [Decision Letter · Decision Letter 0]

14 Nov 2022

Dear Dr Jones,

Thank you very much for submitting your manuscript "TriTrypDB: an integrated functional genomics resource for kinetoplastida" for consideration at PLOS Neglected Tropical Diseases. As with all papers reviewed by the journal, your manuscript was reviewed by members of the editorial board and by several independent reviewers. The reviewers appreciated the attention to an important topic. Based on the reviews, we are likely to accept this manuscript for publication, providing that you modify the manuscript according to the review recommendations. 

Sincerely,

João L. Reis-Cunha, Ph.D.

Academic Editor

Ricardo Fujiwara

Section Editor

Reviewer's Responses to Questions

**Key Review Criteria Required for Acceptance?**

**Methods**

-Are the objectives of the study clearly articulated with a clear testable hypothesis stated?

-Is the study design appropriate to address the stated objectives?

-Is the population clearly described and appropriate for the hypothesis being tested?

-Is the sample size sufficient to ensure adequate power to address the hypothesis being tested?

-Were correct statistical analysis used to support conclusions?

-Are there concerns about ethical or regulatory requirements being met?

Reviewer #1: (No Response)

Reviewer #2: This describes a database resource, so this section is not particularly relevant. Please see my general comments below.

Reviewer #3: (No Response)

**Results**

-Does the analysis presented match the analysis plan?

-Are the results clearly and completely presented?

-Are the figures (Tables, Images) of sufficient quality for clarity?

Reviewer #1: (No Response)

Reviewer #2: Again, this paper describes a database resource, so this section is not particularly relevant. Please see my general comments below.

Reviewer #3: (No Response)

**Conclusions**

-Are the conclusions supported by the data presented?

-Are the limitations of analysis clearly described?

-Do the authors discuss how these data can be helpful to advance our understanding of the topic under study?

-Is public health relevance addressed?

Reviewer #1: (No Response)

Reviewer #2: This paper describes TriTrypDB, a sub-component of the larger VEuPathDB project, which is an excellent and vital resource for research into these neglected tropical pathogens. The paper describes the great range of primary and derived data resources, the powerful tools for searching/interrogating them, and the diverse range of work they support. If anything, I feel the impressive first discussion paragraph undersells the fundamental importance of this resource to the field!

Reviewer #3: (No Response)

**Editorial and Data Presentation Modifications?**

Reviewer #1: (No Response)

Reviewer #2: Overall, the quality of the text and figures are high. Some descriptions of the data underlying the TriTrypDB database could, however be a little clearer - see general comments below. Some minor changes to the text should clarify these points.

Reviewer #3: (No Response)

**Summary and General Comments**

Reviewer #1: This is a nice overview of the TriTrypDB resource. I have a few minor comments. 

- The only part I found confusing was the section on the Tritryp genomes that used to be hosted in GeneDB: "The large-scale re-annotation and curation of these genomes were made possible by continuous

funding from the Wellcome Trust for the TriTrypDB database since 2012. The updated annotations are regularly integrated into TriTrypDB through bimonthly releases. As GeneDB was taken offline last year, TriTrypDB serves as the sole authoritative resource for the annotation of these genomes. " What large scale re-annotation is being referred to? Were they annotated independently or were annotations pulled from GeneDB? Is the sentence about updated annotations referring to before or after GeneDB was taken offline? I think it's great to mention that TriTrypDB is now the sole source, but I was tripped up by the rest of the paragraph.

- Under Record Pages: I felt that the mention of other TriTrypDB pages was a little superficial. "TriTrypDB also provides pages for other record types including popset isolate sequences, genomic sequences (scaffolds), genomic segments, SNPs, ESTs, metabolic pathways and chemical compounds." As someone who hasn't accessed these pages I wondered what they contained/how they were organized/how you get there. This might be difficult to do because the pages contain such different types of data, but if possible a bit more information might help a reader who is less familiar with these pages.

- "In addition, TriTrypDB collaborates with the Gene Ontology Consortium to create new terms to represent kinetoplastid biology, particularly cellular components and biological processes. " This is interesting and I wasn't aware of it. Really minor suggestion, but it could be interesting to mention what some of these terms are!

Under the record pages section there is a typo: "Gene pages host [a] multitude of data in.."

Reviewer #2: I have some comments and queries. However, I would like to emphasise that these are intend as constructive criticism and desirable changes on the large scale - perhaps outside of the scope of this paper. The core of this paper is already strong, a clear description of a resource me, my lab, and my field use extensively and daily.

I find the description of the impressive range of data/analyses that TriTrypDB enables access to clear. However I am afraid I am left a little confused by what TriTrypDB is NOT responsible for, especially at the foundational level, eg.:

Curated genomes. Is 'ownership' of these limited to hosting the sequence and carrying out annotation, or will efforts be made to improve assembly based on new data? 

Genome annotation. Does TriTrypDB carry out proactive gene model updates for curated genomes, updated/corrected gene models for newly submitted genomes, de novo annotation of gene models for new genomes, etc. or must these be provided by the community for new genomes?

Overall, this is a question of defined scope and I suggest making distinctions between "owned" sequence data that may change, "hosted" sequence data that will not change and "live" data which is recalculated each release, and to think where else similar distinctions may help for other data types.

I hope that more clarity here would help the community effectively feed data into TriTrypDB and vice versa.

I found it clear that TriTrypDB draws data from NCBI genomes and SRA, however I found the inverse unclear. Eg. Do updates to the reference genome annotations (particularly gene) get fed back to the NCBI gff files for those genomes? Similarly for community-sourced annotation updates for non-reference genomes? Are all genomes on TriTrypDB also in NCBI genomes?

I realise that good open data practice for the authors of any new genome to appropriately deposit the raw and assembled sequence data into appropriate repositories, but this is not always done - particularly (it seems) for gene models. I suspect this has knockon effects, eg. are all predicted protein sequences available to Uniprot, gathered for UniRef, and similar?

On a related note, you indicate TriTrypDB now includes AlphaFold predictions, but it was unclear whether these are the EMBL-EBI predictions or custom ones? If they are from EMBL-EBI, I believe these use the Uniprot gene models/protein sequences - are these identical to TriTrypDB? 

The update schedule is impressive, with the bimonthly updates, and regular version releases. While this versioning is clear, I find it unclear exactly what is versioned in each release, eg. Most genome sequences are identical between TriTrypDB versions, while some gene models and almost all downstream bioinformatic predictions may not. Some guidance on how to stably refer to analyses would be useful, eg. For a statement like "the L. Infantum ortholog of Tb927.4.1310 is LINF_340041500" should it cite the specific TriTrypDB version that was used?

Old versions of gene models are available, are old versions of the wider database?

On a related note, the community uses gene ID as a stable reference ID, however it is not clear how stable these are. In T. brucei, for example:

Tb927.9.15020: Changed start codon.

How is versioning of such changes shown? I could not find this information, and it could easily trip up reanalysis of older data. 

Genes can also be removed without trace, eg:

Tb927.1.3970: Removed gene model.

I understand that it is important to remove dubious genes which lack any information that they are "real", but older transcriptomic and proteomic data will certainly include such dead gene IDs which are hard to back-trace. 

The origin of GO term data is unclear - some indication of the workflow of assigning predicted and experimentally shown GO classes would be useful.

Better mobile device support would be very desirable, though I recognise that this would be an enormous investment. I suspect this would be especially for the users in countries directly affected by these diseases where mobile infrastructure is often better. Some minor changes, eg. to gene pages when the left index bar is open, could go a long way. 

After making all these criticisms, I would like to finish with a few clear new positives:

Organism preference is a great little feature.

The search pipeline is still very powerful and useful. 

The Apollo tool looks powerful and I look forward to trying it out. 

Similarly, the single cell seq data visualisation is valuable and will be interesting to explore. 

Community integration and annotation of curated genomes are still overall excellent.

Reviewer #3: The submitted manuscript describes several of the key functions of TriTrypDB, the main resource for functional data mining in Kinetoplastid research. The power of TriTrypDB lies in the fact that it not only serves as a collection of datasets but that it allows the clever integration of hundreds of datasets, thus enabling the easy extraction of relevant information. It is impressive how TriTryp has been able to keep up with the ever-increasing generation of NGS, mass-spectrometry and microscopy data over the last decade and managed to incorporate them into one well-functional database. We are thankful for the many new additions, especially the alpha-fold predictions, tools to visualize sc-RNA-seq data and the incorporation of the TrypTag data.

We also appreciate how the TriTryp team fosters the participation of the community to continually improve the quality of the entries, either by enabling the addition of user comments, by using the integrated curation tool Apollo to correct/add annotations and by offering workshops on how to use the database.

Our lab works with TriTrypDB on a daily basis and is truly thankful for the efforts the authors have put into developing it. In addition, we appreciate the TriTrypDB team listening to user suggestions on how to improve access to the currently available data and to add new features.

I have reviewed the manuscript jointly with my postdoc Raúl Cosentino and both of us would be happy to see the manuscript published as it is, something we have never written before. However, as always there are some suggestions we would like to make.

Improvements to the database

• We were surprised to see that there are 17 genomes lacking any form of annotation. Given that a ‘draft’ genome annotation based on sequence homology can be easily done using tools like Companion, wouldn’t it be worth adding it?

• While the community annotation from Apollo is certainly a great feature, at times it can be confusing/misleading. Looking at a region on chr 2 of the T. b 927 genome in the ‘gene models’ category, we see many annotations from nanopore direct RNA sequencing that do not match any of the genes in the reference, e.g. KS17gene_3200a, KS17gene_3201a. https://tritrypdb.org/tritrypdb/app/record/gene/Tb927.10.15350#GeneModelGbrowseUrl

We acknowledge that such annotations are unavoidable with un-curated community annotations, but maybe it would be helpful to clearly list the authors of such annotations. Here the author is listed as api@local.host, which does not tell us much.

• Why are the “Origins of Replication” datasets listed in the category of “Structural variation”? We would associate “Structural variations” with large insertions/deletions or chromosomal rearrangements. Given the transient nature of “Origins of Replication”, maybe they would better fit in the “Epigenomics” category.

• Regarding the “Search strategy system” the authors write, “Saving a strategy retains the order of steps and parameter values, but not the actual results as subsequent database versions containing new data may alter the results”. To increase reproducibility, it would be great if users could store (and share) which search strategy and which specific database they used. This would allow other users to reproduce the results. Such a feature would probably require the option to choose a database version in which to run a search strategy. Maybe it would be useful to implement such an option.

• Running the big-PI predictor for a gene we received the following error message: “Can't connect to mendel.imp.ac.at:443 (certificate verify failed) SSL connect attempt failed with unknown error error:14090086:SSL routines:ssl3_get_server_certificate:certificate verify failed at /usr/share/perl5/LWP/Protocol/http.pm line 51.”

We are aware that the predictor is an external tool but maybe the authors could check to make sure that this was just a temporary error message.

• Given that the majority of Trypanosome brucei research is done with the lister 427 strain, it would be great if it could be added as an additional curated reference strain.

Improvements to the manuscript

• It would be interesting to see how the number of datasets incorporated into TriTrypDB has increased over the last decade. Would it be possible to plot this? Such a plot would also nicely illustrate to the reader the amount of data available in TriTrypDB.

• In the “Future perspectives” section, it may be worth mentioning the prospect of working with phased diploid genomes and the challenges this could cause. 

In summary, we would like to thank the authors for the development and maintenance of an excellent database.

Raúl Cosentino and Nicolai Siegel

PLOS authors have the option to publish the peer review history of their article (what does this mean?). If published, this will include your full peer review and any attached files.

Reviewer #1: No

Reviewer #2: No

Reviewer #3: Yes: Raúl Cosentino and Nicolai Siegel

Figure Files:

Data Requirements:

Reproducibility:

References

---

## [Editor Report · Decision Letter 1]

23 Dec 2022

Dear Dr Jones,

We are pleased to inform you that your manuscript 'TriTrypDB: an integrated functional genomics resource for kinetoplastida' has been provisionally accepted for publication in PLOS Neglected Tropical Diseases.

Best regards,

João L. Reis-Cunha, Ph.D.

Academic Editor

Ricardo Fujiwara

Section Editor

I would like to thank the authors for addressing all the points raised by the reviewers.

I am also a frequent user of TriTrypDB, and I congratulate the authors for their important work.

---

## [Editor Report · Acceptance letter]

15 Jan 2023

Dear Dr Jones,

We are delighted to inform you that your manuscript, "TriTrypDB: an integrated functional genomics resource for kinetoplastida," has been formally accepted for publication in PLOS Neglected Tropical Diseases.

Best regards,

Shaden Kamhawi

co-Editor-in-Chief

Paul Brindley

co-Editor-in-Chief
